# Conserving the Ogallala Aquifer in Southwest Kansas: From the Wells to People, A Holistic Coupled Natural-Human Model

Joseph A. Aistrup[1], Tom Bulatewicz[2], Laszlo J. Kulcsar[3], Jeffrey M. Peterson[4], Stephen M. Welch[5], and David R. Steward[6]

[1]Department of Political Science, Auburn University, Auburn, AL 36849, USA
[2]Department of Computer Science, Kansas State University, Manhattan, KS 66506, USA
[3]Department of Agricultural Economics, Sociology and Education, Pennsylvania State University, State College, PA 16801, USA
[4]Water Resources Center and Department of Applied Economics, University of Minnesota, St. Paul, MN 55108, USA
[5]Department of Agronomy, Kansas State University, Manhattan, KS 66506, USA
[6]Department of Civil Engineering, Kansas State University, Manhattan, KS 66506, USA

*Correspondence to:* D. R. Steward (steward@ksu.edu)

**Abstract.** The impact of water policy on conserving the Ogallala Aquifer in Groundwater Management District 3 (GMD3) in Southwest Kansas is analyzed using a system-level theoretical approach integrating agricultural water and land use patterns, changing climate, economic trends, and population dynamics. In so doing, we 1) model the current hyper-extractive coupled natural-human system (CNH), 2) forecast outcomes of policy scenarios transitioning the current groundwater-based economic system toward more sustainable paths for the social, economic and natural components of the integrated system, and 3) develop public policy options for enhanced conservation while minimizing the economic costs for the region's communities. The findings corroborate previous studies showing that conservation often leads initially to an expansion of irrigation activities. However, we also find that the expanded presence of irrigated acreage reduces the impact of an increasingly dryer climate on the region's economy and creates greater long-term stability in the farming sector along with increased employment and population in the region. On the negative side, conservation lowers the net present value of farmers' current investments and there is not a policy scenario that achieves a truly sustainable solution as defined by Peter H. Gleick. This study reinforces the salience of interdisciplinary linked CNH models to provide policy prescriptions to untangle and address significant environmental policy issues.

## 1 Introduction

Our world faces a public policy conundrum. Crop yields on many varieties have tripled over the past 50 years, with irrigated cropping practices accounting for 40% of the total increased level of production (United Nations, 2011, p.3). Even so, food deserts, often created by market inequities, leave over 1 billion people worldwide malnourished (United Nations, 2011, p.ix). If projections are correct, the situation in the future does not improve. By 2050, the U.N. estimates that the world's population will have grown by another 2.2 billion people, most of whom will live in impoverished countries (Nations, 2017), while demand for crops and meat products will soar by 70% (United Nations, 2011, p.7). Irrigated crops will continue to be crucial for meeting

the world's future demands for nutrition. Unfortunately, many irrigated croplands are in regions that are, or are becoming, fresh water challenged (United Nations, 2011, pp.7-12).

This is the case for the High Plains Aquifer in the United States, also referred to as the"Ogallala Aquifer" (see Figure 1) The entire aquifer spans 450,000km$^2$ and underlies 27% of the irrigated land in the United States (Dennehy, 2000). Powell (1879) classifies this semi-arid grassland ecosystem as an arid land lying west of the 100th meridian with a mean annual precipitation less than 500mm (20in). The aquifer is one of four "critical areas" for "annual renewable water" in the Western Hemisphere and one of 22 worldwide (Montaigne, 2002).

Thus far, the scientific community has developed hydrological models that confirm what we already know; that this natural system aquifer is already past its peak groundwater depletion (Steward and Allen, 2016) and will soon be so diminished that it can no longer sustain its current irrigation farming practices (McGuire, 2014; Scanlon et al., 2012; Steward et al., 2013). This policy study uses a coupled natural-human system (CNH) approach focused on Groundwater Management District 3 in Southwest Kansas. The 12 counties of GMD3 are classified as hyper-extractive (Aistrup et al., 2013), with similar water resource extraction patterns as other Ogallala counties. To study GMD3, we develop an integrated, cross-disciplinary, system-level, theoretical approach, linking agricultural land and water use practices, changing climate patterns, economic trends, and population dynamics to issues of groundwater sustainability. In so doing, we 1) accurately model the current CNH system, 2) forecast the outcomes of policy scenarios to transition the current groundwater-based economic system toward avenues that are more sustainable for the social, economic and natural systems, and 3) develop public policy options the conserve the aquifer while minimizing the economic cost for the region's communities.

## 2  Methods

The scope of this study is the 12 counties of Groundwater Management District #3 (GMD3) in Southwest Kansas (Figure 1). Farmers in this region of the Ogallala, similar to other regions of this aquifer, tap groundwater to raise corn, sorghum, soybeans, wheat and alfalfa. In turn, the irrigated grains and alfalfa supply inputs for large-scale confined feedlots that deliver finished cattle for several of the world's largest meatpacking factories (Broadway and Stull, 2005), making GMD3 one of the most productive value-added agricultural regions in the U.S. and world (Steward et al., 2013). However, as illustrated in Figure 1, the emergence of this value-added agricultural economy has imposed a heavy toll on the aquifer, reducing its saturated thickness and altering its recharge (Custodio, 2002). With irrigated crops accounting for 97% of the water withdrawals from the Ogallala Aquifer in Western Kansas (United States Department of Agriculture, 2013), developing policies that conserve the life of the aquifer is essential for maintaining the long-term economic health and vitality of the region and Kansas.

Our framework for studying GMD3 is a coupled natural-human system approach. CNH studies 1) take advantage of both social and natural systems variables, 2) are multidisciplinary in theoretical approach, 3) integrate research methods across disciplines, and 4) are "context specific" while understanding temporal dynamics (Liu et al., 2007). Within CNH studies, our framework fits under a class of hydro-economic models that Brouwer and Hofkes (Brouwer and Hofkes, 2008) identify as

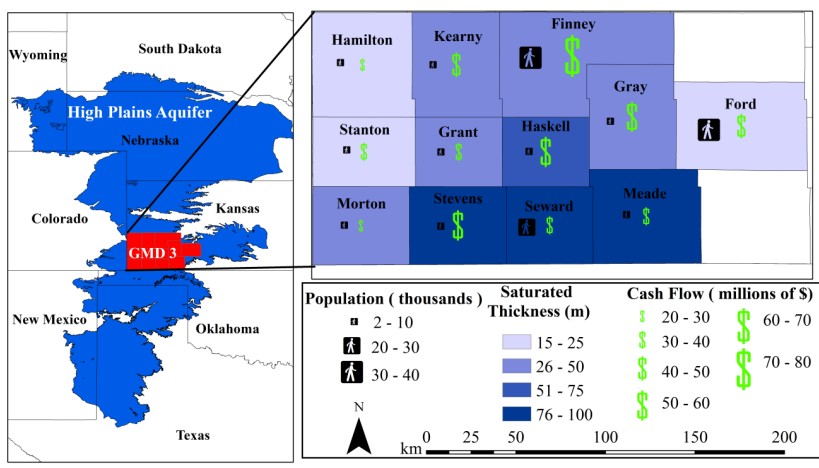

**Figure 1.** The Groundwater, Population, and Cash Flow Summary Statistics for the 12 Counties of Groundwater Management District 3. Created by Weston Koehn.

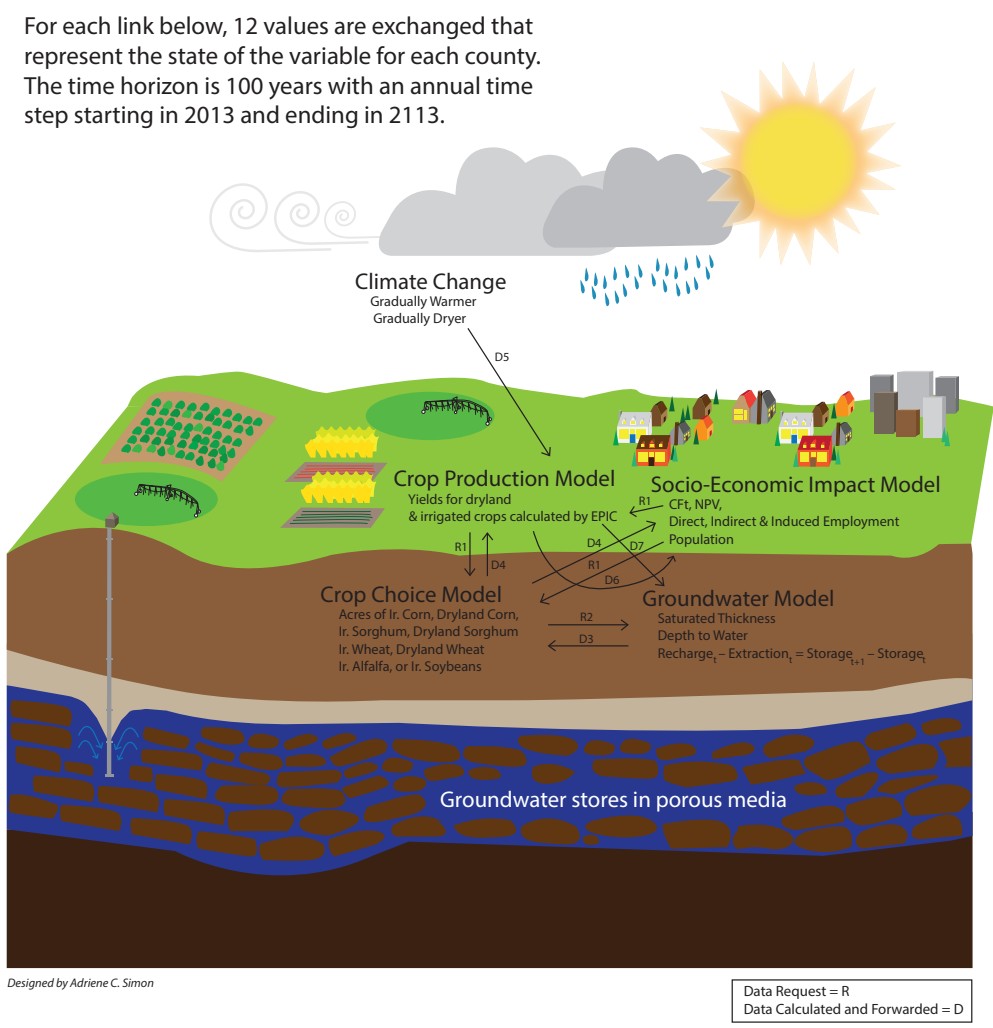

For each link below, 12 values are exchanged that represent the state of the variable for each county. The time horizon is 100 years with an annual time step starting in 2013 and ending in 2113.

Climate Change
Gradually Warmer
Gradually Dryer

D5

Crop Production Model
Yields for dryland
& irrigated crops calculated by EPIC

Socio-Economic Impact Model
CFt, NPV,
Direct, Indirect & Induced Employment
Population

R1

R1

D4

D4      D7

R1

D6

Crop Choice Model
Acres of Ir. Corn, Dryland Corn,
Ir. Sorghum, Dryland Sorghum
Ir. Wheat, Dryland Wheat
Ir. Alfalfa, or Ir. Soybeans

R2

D3

Groundwater Model
Saturated Thickness
Depth to Water
Recharge$_t$ – Extraction$_t$ = Storage$_{t+1}$ – Storage$_t$

Groundwater stores in porous media

Designed by Adriene C. Simon

Data Request = R
Data Calculated and Forwarded = D

**Figure 2.** CNH Model Components.

modular (Volk et al., 2008; Jonkman et al., 2008; Barton et al., 2008), and holistic (Ward and Pulido-Velazquez, 2008a; Cai et al., 2008; Pulido-Velazquez et al., 2008).

## 2.1   Integration Methodology:

The integrated model is composed of independent disciplinary models that each conform to and are linked within the Open Modeling Interface (OpenMI) Standard (Gregersen et al., 2007). The linked model consists of four components with the input-output mappings shown in Figure 2 with one exogenous component, climate. The lowest common unit of analysis across all modeling components operates at the county level, of which there are 12 in GMD3, and the models conduct their simulations

and exchange data over this set of 12 polygons. The components operate on an annual time step over the 100-year horizon from 2013 to 2112. For each year of the simulation, the socio-economic impact model requests (see R1 in Figure 2) the crop acreage from the crop choice model and crop yield from the crop production model, which also requests the crop acreage from the crop choice model. The crop choice model in turn requests (R2) the saturated thickness and depth to water from the groundwater model (for the previous year), which triggers the groundwater model to simulate the previous year and provide the data (D3). The crop choice model then calculates the acreage data for the current year and provides them (D4) to the crop production and socio-economic impact models. The crop production model then calculates the yields and water use (D5) for the year and provides them to the socio-economic impact model (D6) and groundwater model (D7), respectively, allowing them to calculate their outputs for the year.

## 2.2 Socio-Economic Impact Model

The socio-economic impact model uses Cash Flow at time $t$ ($CF_t$), also referred to as "Gross Profit." $CF_t$ = Revenue (i.e. Price*Yield) - Costs (Fixed Costs + Variable Costs + Lift Costs). Based on $CF_t$, the Net Present Value (NPV) is the discounted $CF_t$, calculated as $\Sigma$ ($CF_t/(1+K_t))^t$ where $K$ is the discount rate on investments (set at 4%) and $t$ is a counter for the year, starting with 1 in 2013. In this model and in the crop choice model, all monetary variables are in inflation-adjusted 2013 dollars. Many economic impact models also estimate economic multiplier effects using Impact Analysis for Planning (IMPLAN) (Group, 2010). IMPLAN produces direct, indirect and induced impacts on total economic activity, value-added activity, and employment. Our linked CNH model focuses on community impacts that yield employment opportunities to support a stable population base. Over the years, as agricultural production has become increasingly mechanized and technologically based, farms have consolidated, creating larger farms with fewer employees (Flora et al., 1992). This has led to considerable population decline in most rural farming dependent communities in the Great Plains region. Even though wealth creation from farming is important, from a community development standpoint, this wealth is most important when it translates to jobs and population. In Western Kansas, IMPLAN estimates that it takes about $1 million $CF_t$ to produce 1.2 full-time equivalent (FTE) employees, of which .88 FTE are directly tied to crop production agriculture and .32 FTE represent indirect and induced employment. There are slight variations between $CF_t$ for irrigated cropland and non-irrigated cropland, which are detailed in Table A1 in the Supplementary Materials.

To estimate the population impact of this employment, we conducted a cross sectional time series analysis with panel corrected standard errors (TSCS) (Beck and Katz, 1995, 2011) from 1970 to 2010, where the time increments are every 5 years (1970, 1975, 1980 etc.), the cross sections are the 12 counties of GMD3, and the error terms are both heteroskedastic and serially correlated (AR1).

$$y_{i,t} = \beta_n x_{i,t} + \epsilon_{i,t} \tag{1}$$

Where: $\epsilon_{i,t} = \upsilon_{i,t} + \rho\epsilon_{i,t-1}$. In this 40 year TSCS regression model, the independent variables are each county's employment in levels (# of employees) in agriculture, manufacturing, construction, health care, government services, and education, while

the dependent variable is each county's total population. We obtained U.S. Census population data and Bureau of Economic Analysis employment data from Woods and Poole (Woods and Economics, 2012). This equation explained 92% of the variance in total population in these counties during this time period (See Supplementary Materials for model results). We multiplied the regression coefficient for agricultural employment (2.15) by the IMPLAN's employment multiplier to calculate the impact on population (2.15*(.88)= 1.88 people). Thus, each \$1 million $CF_t$ from crop production in GMD3, supports an additional .88 FTE and 1.88 residents in region. Given the connection between agriculture and value-added meat production, we assign the other .32 FTE emanating from indirect and induced employment impacts to the coefficient for manufacturing (.32*1.33 = .43 people). Taken together this suggests that each \$1 million in $CF_t$ from crop production supports 1.2 additional jobs and 2.3 more people living in the region.

## 2.3 Crop Choice Model:

The crop choice component is an iterative Positive Mathematical Programming (PMP) model (Howitt, 1995) that simulates farmers' allocation of arable land to different crops in each county. The model operates on an annual time step, with each execution predicting farmers' choices in a single growing season.[1] In addition to harvested crop prices and crop-specific costs of production, the model accepts as inputs the current (county average) depth to water and saturated thickness of the aquifer. Depth to water affects water extraction costs, while saturated thickness affects the pumping rate of wells, which in turn creates an upper bound on the annual extraction of irrigation water. Recent work has emphasized the role of well pumping rates on crop and water use choices (e.g., Foster et al., 2015). The model simulates land allocations as the solution to a constrained optimization problem that represents farmers' profit-maximizing mix of land uses, given price conditions, water extraction costs, and the constraints on water and land availability. The model outputs are the predicted acres planted to each crop.

A separate instance of the model was calibrated to data from each county in the study region. Each county model simulated acreages for irrigated and nonirrigated plantings of the five dominant in crops in the region: wheat, corn, sorghum, soybeans, and alfalfa. Nonirrigated production of soybean and alfalfa is unfeasible given regional hydroclimatology and so are only included as irrigated crops. Thus, eight crop categories were modeled. The models were calibrated to the 2006-08 average of observed acreages, yields, and prices for the eight crops by county, the most recent period for which comprehensive county-level data are available from the National Agricultural Statistics Service (NASS). Expected crop yields are simulated within the model from water response functions in Martin (1984), which were calibrated to yield data from NASS and weather data from the Kansas Weather Data Library.[2]

There is a long history of increasing crop yields due to genetic improvements in plant varieties (United Nations, 2011, p. 46). Considering this history and the continued investment in plant genetics by industry and governmental agencies, the crop choice model assumes that yields will continue to improve into the future based on the noncompunded percentage growth

---

[1]An annual horizon reflects the fact that farmers competitively extract water from a common pool, leaving no individual incentive to conserve stocks that can be withdrawn by other users users in future periods (Koundouri, 2004).

[2]As noted above, the crop choice model is not calibrated to a specific year, but rather to the mean outcome during the base period. Calibrating to a specific year would "overfit" the model so that it replicates that single year but doing a poor job with the years before or after that one year. The mean approach does not give an exact fit for any single year but makes the model match better with the data cloud.

rates estimated from time series regression of Western Kansas yields for irrigated and non-irrigated plant varieties from 1974 to 2009 (see Supplementary Materials) (Rogers and Lamm, 2012). The average annual improvements in yield range from a high of 1.28% for irrigated corn, to a low of .55% for dryland wheat and .53% for irrigated sorghum. Crop production costs, excluding irrigation, were obtained from Kansas State University Extension enterprise budgets and were increased over time at noncompounded percentage rates in proportion to yields based on a regression analysis of budget and yield data using 2006-12 observations. While base level yields and costs were calibrated to 2006-08 data, growth percentages were applied to adjust the initial simulation year to correspond to 2013. The cost of irrigation water was calculated separately based on the energy costs from pumping lifts (Rogers and Alam, 1999). Details on the model development and calibration methods are in Clark (2009), Bulatewicz (2014), and Armoa (2015).

## 2.4 Crop Production Model:

The crop production component projects grain yield and irrigated water by using the Erosion-Productivity Impact Calculator (EPIC) model (Williams, 1995). EPIC simulates daily crop growth by representing three major processes: (1) phenological development; (2) dry matter production and partitioning to plant tissues resulting in growth; and (3) economic yield. The model reproduces the results of irrigation, fertilization, tillage, variety selection, alternative production calendars, etc. Plant growth is estimated from intercepted solar energy and plant leaf area. Daily dry matter is accumulated for the growing season as controlled by heat units or environmental conditions (typically freeze events for summer crops) and yield is estimated using a total biomass to grain ratio. EPIC is able to simulate multiple crops because it embodies a generic plant model that can be parameterized to represent different species.

We previously calibrated the model for use in western Kansas (Bulatewicz et al., 2009) and further refined the parameters to support non-irrigated cropping for this study. The model component, developed in an earlier effort (Bulatewicz et al., 2014) and implemented using the Simple Model Wrapper (Castronova and Goodall, 2010), has an embedded set of simulated output data from EPIC collected by executing the model for all combinations of the relevant inputs (soil, crop, management, weather). The component operates over a set of (independent) polygons of variable size, accepting inputs for soil type, weather station, and crop, and providing outputs for yield and water-use.

## 2.5 Climate:

Each simulated year's weather is determined by a random draw from meteorological records between 1985 and 2012. We build the likelihood of climate change into our simulation about the future of GMD3. ECHAM5 climate change models for the High Plains region suggest future regional warming and a gradual increase in extreme weather events, pointing toward a less suitable climate and thus reduced yields for agricultural production (Zabel et al., 2014). To model this climatic progression, we weight the weather data from 1985 to 2012 such that years of below-average dryness will, over 100 years, gradually become 25% more

likely to occur than they are now. We find that this captures both the prolonged periods of drought that are likely to become typical, while allowing for shorter-lived periods of plentiful precipitation.[3]

## 2.6 Groundwater Model:

The groundwater modeling component provides estimates of groundwater storage and the changes in storage due to pumping and natural hydrologic processes. This model is linked to the crop production and economic crop choice model using OpenMI (Gregersen et al., 2007), and operates at the common county level scale. Conservation of mass requires that recharge minus extractions is equal to the annual change in storage:

$$\text{Recharge}_t - \text{Extraction}_t = \text{Storage}_{t+1} - \text{Storage}_t \tag{2}$$

This groundwater model integrates these spatially and temporally variable components of the hydrologic cycle to provide these fluxes at the common county-level aggregation scale, which are consistent with the scales of previous studies in the study region (Steward and Allen, 2013, 2016).

Specific steps used to prepare groundwater data follow. Storage is obtained from groundwater observation wells, kriging across wells to give a surface of saturated thickness, multiplying by specific yield to give water content, and integrating across the aquifer area within a county (Steward and Allen, 2013). Recharge is obtained by spatially integrating results from Hansen (1987),[4] and extraction is obtained from the crop irrigation component, which was parameterized against historical pumping rates recorded in the WRIS water-use reports. The county level model ignores the changes in storage resulting from groundwater movement between counties, since groundwater moves with an average velocity of only 30 cm/day driven consistently by the west-east sloping aquifer (Gutentag et al., 1984), and the differences between what enters and leaves each county represents a very small fraction of changes in storage. This county level model provides groundwater availability for the crop production model, and also ignores other water use such as municipalities, industry and feedlots, which have historically used much less than 5% of the groundwater extractions in each county.[5]

The baseline model predictions accurately reproduced the groundwater data throughout the historical period. Model results were also compared to the future predictions of a higher-resolution fishnet model of Seward County (Steward et al., 2009).

---

[3]We compared the statistics of the original 27 years of weather data the our 100 year simulation where the dryer years gradually became 25% more likely. The average maximum temperature increased from 20.09C to 20.14C, the average total precipitation declined from 477.83mm to 458.58mm, while the relative humidity declined from 63.95 to 63.63.

[4]It is important to note that the recharge component is kept consistent throughout this study. Given the relatively thick soil units throughout much of the region (Gutentag et al., 1984), recharge rates are low and may take decades or longer to reach saturated groundwater (McMahon et al., 2007). This is consistent with Sophocleous (2005) who noted that groundwater pumping usually has little impact on the recharge.

[5]In the simulation, groundwater pumping is based upon an annual decision for when to start pumping and when to stop, where the well is traditionally left on throughout the growing season. Thus the dynamics are drawdown throughout a growing season and recovery before the next pumping cycle, where large drawdowns may occur when the wells actively pump (Mullican III, 2012), and a new elevation becomes established during a recovery period (Dugan et al., 1994, 23). Thus, the groundwater model exchanges data at the same annual frequency as the economic crop choice model that dictates the annual water requirements.

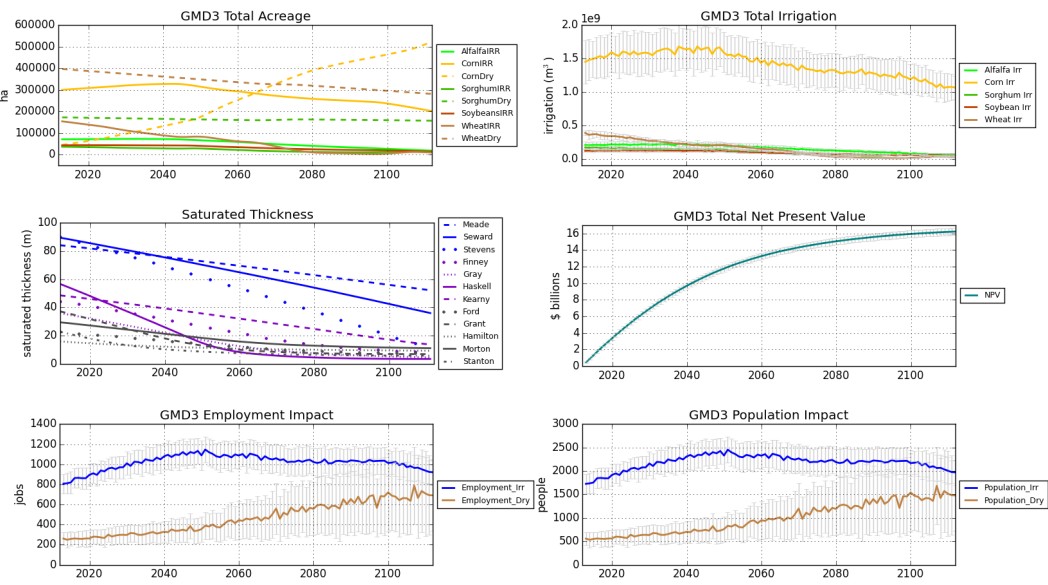

**Figure 3.** Baseline Outcomes of Current Irrigation Practices in GMD3: 100 simulations. Top Row: The left graph shows the acreage planted for dryland and irrigated crop varieties. Note the demise of irrigated corn and the rise of dryland corn over the course of the simulation.⁂ The right graph depicts total water consumed in GMD3 by each irrigated crop. Middle Row: The left graph shows the average level of saturated thickness of the Ogallala in each of the 12 counties over time. Note that only two of the 12 counties end the simulation with an average saturated thickness greater than 15m. Generally, more than 15m of saturated thickness is need to irrigate using center pivots.⁂ The right graph sums the NPV across all 12 counties over time. Bottom Row: The left graph illustrates the predicted number of additional jobs created in a simulated year due to the direct, indirect and induced impacts of irrigation and dryland crop production, respectively. Note that over time, the level of variation in employment due to dryland crop production is exceptionally large, reflecting the greater levels of uncertainty introduced by a gradually warmer climate. The graph on the right depicts the number of additional people who live in the GMD3 in that year because of irrigated and dryland crop production. The population impact mirrors the level of variation shown in the previous graph. ⁂ The whiskers denoting the level of uncertainty ($2\sigma$) are excluded on this graph to maintain the clarity of presentation.

Similarly, the results from the model resolution of this study, when aggregated to the region, reproduce longer term regional projections (Steward et al., 2013).

## 3  Baseline, Point Estimates and Estimates of Uncertainty

This integrated model is used to develop point estimates for important variables and also estimates of uncertainty by simulating a policy scenario 100 times, where each policy scenario simulated a period of 100 years, and that the parameter that was re-sampled each simulation was the weather-year. Figure 3 illustrates the significant findings from the groundwater, crop choice,

crop production and socio-economic impact models. The point estimates for each component of each year is presented as the average from the 100 simulations and the estimates of uncertainty are the standard deviations around each point estimate.

Our holistic CNH model predicts an unsustainable outcome for the aquifer in all counties if current conditions remain unabated, which is consistent with similar results obtained using different methodological approaches (McGuire, 2014; Scanlon
et al., 2012; Steward et al., 2013; Steward and Allen, 2016). Results in figure 3 illustrate that the acreage for irrigated corn continues to increase until 2030 (ca. 328K ha), showing that corn will remain profitable despite increased pumping depths. However, as the average saturated thickness in counties falls below 9m, the saturated thickness necessary for full use of center pivot irrigation (Hecox et al., 2002),[6] in 6 of 12 counties, and below 15m in all but Meade and Seward, the amount of acreage planted in irrigated corn is projected to decline significantly. By 2112, there are fewer than 200K ha of irrigated corn. On the
other hand, the acres planted to dryland corn soars to 500K ha, surpassing by 2070 the acreage planted in dryland wheat. As the capacity of the aquifer to support irrigation decreases over time, the average saturated thickness in each county stabilizes at less than 10m, reflecting that, average, farmers no longer have the capacity to consistently draw the high volumes of water necessary for center pivot irrigation Even though the $CF_t$ from irrigated crops will increase from \$400M to just over \$600M, the $CF_t$ from dryland crops will increase from \$50M at the beginning of the simulation to about \$400M at the end of the
simulation. This leads to continued positive employment impacts for irrigated and dryland agriculture, increasing from a total of 1,000 workers currently to 1,800 workers at the end of the simulation. The NPV reaches \$10B by 2040 and \$13.3B by 2060.

However, hidden behind these point estimates is much uncertainty that comes from relying on dryland farming in the semi-arid counties of GMD3. This is the most apparent by examining the outputs of the socio-economic model. Although the $CF_t$ for all dryland production is generally positive, the variation around that average grows (whiskers represent $2\sigma$'s from the average)
over time relative to the increased number of acres planted to dryland crops and the increased prevalence of a dryer climate. This creates a higher probability for dryland crop failure in years with lower rainfall levels.

We have two points of caution regarding these findings. First, the accuracy of as any model of this nature declines as it forecasts into the future. Thus, some of this uncertainty in dryland crop production is a function of our modeling methodology. Second, currently, farmers in GMD3 and U.S. mitigate the risk from drought and other weather related calamities with federally
subsidized crop insurance, administered by the Risk Management Agency of the USDA. In the event of crop failure, insured farmers receive a settlement based primarily on the price of the crop during harvest multiplied by the average yield in that county over the past 10 years (Risk Management Agency of the United States Department of Agriculture, 2016). In practice, crop insurance payments for farmers are much less than the cash flow from that crop in an average year. This suggests that as the climate becomes hotter and dryer, the negative swings in employment and population will be significant even if a portion of
farmers' incomes from dryland fields are insured. This also reinforces an important historical lesson. In a retrospective analysis of counties in the Great Plains, Hornbeck and Keskin (2014) found that new access to groundwater for irrigation mitigated the impact of drought. However, drought sensitivity in irrigated counties began to increase as farmers switched to higher value water-intensive crops and groundwater access declined.

---

[6]Even though the average saturated thickness for most counties is low, there is much variation among wells in each county. Thus, some wells may have more than 9m of saturated thickness, allowing these farmers enough water to irrigate (Hecox et al., 2002), while others may have less.

**Table 1.** Summary of Major Outcomes from each Scenario

(Standard Deviations in Parentheses)

| Scenario | Status Quo | 1st | 2nd | 3rd | 4th | 5th |
|---|---|---|---|---|---|---|
| Description | Baseline | Jr. Rights | 2X Interval | 3X Interval | 4X Interval | 6X Interval |
| Water Reduction | 0% | 21% | 12% | 26% | 35% | 48% |
| #Counties <9m | 6 | 4 | 5 | 2 | 2 | 0 |
| #Counties <15m | 10 | 7 | 8 | 6 | 6 | 2 |
| AvgSatThick GMD3 2013=56.7m, in 2112: | 14.1m | 23.8m | 20.0m | 26.2m | 30.2m | 36.3m |
| ha Irrig. Corn 2013 | 300K | 282K | 300K | 300K | 300K | 300K |
| | (157) | (75) | (84) | (62) | (47) | (55) |
| ha Dry Corn 2013 | 45K | 151K | 45K | 45K | 45K | 45K |
| | (239) | (52) | (129) | (95) | (72) | (84) |
| ha Max Ir.Corn/Year | 328K/2043 | 329K/2063 | 350K/2055 | 376K/2071 | 400K/2083 | 425K/2111 |
| ha Irrig. Corn 2112 | 200K | 270K | 274K | 336K | 370K | 425K |
| | (6.7K) | (2.5K) | (2.6K) | (2.3K) | (2.2K) | (1.3K) |
| ha Dry Corn 2112 | 522K | 452K | 442K | 348K | 395K | 235K |
| | (10K) | (2.9K) | (3.9K) | (3.6K) | (3.0K) | (.64K) |
| NPV 2060 in Billions | $13.3 | $11.3 | $12.1 | $10.6 | $9.5 | $7.6 |
| | ($.381) | ($.526) | ($.505) | ($.542) | ($.612) | ($.608) |
| NPV 2112 in Billions | $16 | $14.5 | $15.3 | $13.7 | $12.5 | $10.1 |
| | ($.430) | ($.567) | ($.515) | ($.552) | ($.690) | ($.607) |
| # Employ 2013 | 1060 | 975 | 980 | 935 | 915 | 835 |
| | (120) | (143) | (138) | (163) | (160) | (172) |
| Population Impact 2013 | 2300 | 2100 | 2100 | 2000 | 1950 | 1780 |
| | (255) | (305) | (295) | (162) | (342) | (172) |
| # Employ 2112 | 1600 | 1840 | 1900 | 1940 | 1880 | 1655 |
| | (355) | (445) | (495) | (463) | (492) | (595) |
| Population Impact 2112 | 3450 | 3930 | 4065 | 4150 | 4025 | 3550 |
| | (760) | (953) | (1059) | (990) | (1055) | (1274) |

## 4  Sustainability and the Ogallala in GMD3

Our definition of sustainability parallels that of Peter H. Gleick, who defines sustainability in terms of using water to allow "human society to endure and flourish into the indefinite future without undermining the integrity of the hydrological cycle or the ecological systems that depend on it." (Gleick, 2000) We add, however, one proviso to this definition. The economies of
the High Plains Aquifer region have already substantially depleted the aquifer (see Figure 1). Stream flows for riparian and aquatic ecosystems have been impaired (Ahring and Steward, 2012). Reversing the impacts of the past 50 years may not be possible without ceasing all irrigation activity in the region. Even then, given the recharge rates of the aquifer, it would take 500 to 1,300 hundred years to fully recharge the aquifer in Western Kansas (Steward et al., 2013). This is not a viable scenario.[7] Thus, our sustainability policy scenarios focus on maintaining current saturated thicknesses and stemming the current pattern
of continuous depletion, while maintaining to the extent possible the employment levels, wealth generation, and population impacts in the region.[8]

### 4.1  Scenarios 1 & 2

The first two policy scenarios use two separate Kansas water conservation statutes to model different policy approaches for achieving a 10% to 20% reduction in irrigation. The Kansas Groundwater Management District Act contains provision K.S.A.
82a-1036, which allows the Chief Engineer to designate an Intensive Groundwater Use Control Area (IGUCA) to implement corrective control provisions reducing the permissible groundwater withdrawal based upon relative dates of priority of such rights (this statute also allows for a rotating schedule(Steward et al., 2008)). Thus, this first scenario takes at least 20% of fields out of irrigated crop production based on senior versus junior water rights. This is modeled by reducing by 20% the acreage assigned to irrigation in each county. We assume that this acreage will be returned to dryland production. The second
policy scenario emanates from K.S.A. 82a-1041(d)(1), which allows adjacent water users in a region to create "Local Enhanced Management Areas" (LEMA). Under this statute, if a large consensus of irrigators in a contiguous area agree to limit water use by a prescribed percentage then that reduction becomes a legally enforceable limitation on all irrigators. Currently, there is one LEMA restricting irrigation in Kansas located in Sheridan and Thomas counties, which are north of GMD3. LEMA's have two advantages over the IGUCA approach. First, it is a bottom-up process, where irrigators in an area agree to the restrictions
through a consensus among them instead of regulations set centrally by the Chief Engineer. Elinor Ostrom's (2010) work suggests that this is a better institutional design for managing common pool resources. Second, this represents a "shared pain" approach; an approach that is preferred by irrigators over the enforcement of junior vs. senior water rights. We operationalize the LEMA policy scenario by assuming that irrigators in GMD3 agree to create a LEMA that doubles the interval between irrigation applications, thus slowing the rate of water application by about 12% across the management district. Based on
previous research, we do not anticipate that either of these scenarios will produce a sustainable outcome. Rather, the focus here is on the effects of restrictions on farmers' NPV and the local communities.

---

[7]Even though some have suggested that it may be possible to import water from the Missouri River basin in South Dakota, or some other river; this solution is both expensive and creates potentially negative environmental consequence for the river from which the water is drawn.

[8]None of the scenarios are constrained to achieve a desired outcome.

Table 1 reports the results of simulations for scenarios 1 and 2 compared to the baseline analysis. Removing 21% of fields from irrigation based on junior water rights means there is an immediate reduction in the number of irrigated acres in all the crop varieties, however irrigated corn only decreases by 18K ha (300K ha in the baseline to 282K ha in junior rights scenario). The Crop Choice Model suggests that the bulk of acres taken out of irrigated production will move to dryland corn (an increase of 106K ha in 2013 over the baseline) production. By contrast, doubling the irrigation interval has little appreciable impact on crop choices in GMD3 compared to the baseline. For scenario 1, the maximum number of acreage for irrigated corn tops out in 2063 at 329K ha compared to 350K in 2055 in scenario 2. Interestingly, under scenario 2, the number of acres under irrigation increases incrementally from 741K ha to 811K ha in 2070 (not shown). This pattern is consistent with previous research (Ward and Pulido-Velazquez, 2008a, b; Steward et al., 2013) noting that farmers use their water savings on more fields to increase their capital returns.[9]

Both scenarios improve the lifespan of the aquifer by 10 to 15 years, but neither comes close to achieving aquifer sustainability given the very slow rate of recharge in most of GMD3. Both approaches produce similar types of outcomes for $CF_t$ and employment. The NPV by 2060 is \$11.3B for the junior rights scenario and \$12.1B for the LEMA scenario, compared to \$13.3B for the baseline. Interestingly, in the long term, communities in GMD3 benefit from conservation. Extending the life of the aquifer in both scenarios leads to more people being employed by the direct and indirect impacts of production farming (an average of 340 to 400 workers a year by 2112).

## 4.2 Scenarios 3, 4, & 5

Given that neither of first two policy options achieves sustainability, we explore the relationship between water conservation and the associated socio-economic consequences for the farmers and communities in GMD3. To do so we simulated the implementation of a LEMA across GMD3 that incrementally increases the interval for irrigation by 3X (a 26% reduction compared to 2014 usage), 4X (a 35% reduction), and 6X (a 48% reduction). We focus on the LEMA policy approach because it is a theoretically more pleasing policy prescription (see arguments above and Ostrom's work (1990)). Significantly, increasing the interval by 6X stretches to the maximum limit of the marginal utility of irrigation for the purpose of assuring increased crop yields. Thus, after the 6X point, the LEMA approach begins to lose its policy integrity.

Tripling the irrigation interval for irrigated corn production gradually increases the acres in production from 300K ha in 2013 to a peak of 376K ha by 2071. Dryland corn, which in the baseline analysis becomes the most predominant crop after 2080, only surpasses irrigated corn in acres planted in 2110, at the end of the 3X simulation. Given the large variation in yields and revenues associated with dryland corn production, policies that reduce dependence on this high risk crop are desirable. The 3X scenario tends to benefit the communities of GMD3 as the number of additional people employed due to the direct and indirect impacts of production agriculture increases from fewer than 1,000 in 2013 to 1,940 in 2112. Similarly, the number of people living in the region because of direct and indirect economic impacts from irrigation and dryland farming increases from 2,000 in 2013 to 4,200 in 2112.

---

[9]Scenarios 3, 4, and 5 follow this same pattern.

Disappointingly, increasing the irrigation interval by 4X or 6X does not produce sustainable outcome for the aquifer. Six of the 12 counties under the 4X scenario and two of 12 counties under the 6X scenario still end the simulation with average saturation depths of 15m or less. There is also an economic cost to irrigators to achieve this level of water savings. The NPV in 2060 shrinks to $9.5B for 4X scenario and $7.6B for the 6X scenario. On the positive side, after an initial decline in employment and population early in the simulation, both rebound to levels just slightly below the other scenarios.

## 5    Conclusions

These results corroborate previous studies that show that conservation often leads initially to an expansion of irrigation activities, as farmers use their water application savings on more fields to increase their capital returns (Ward and Pulido-Velazquez, 2008a, b; Steward et al., 2013). However, our coupled model extends this finding by showing that the expanded presence of irrigated acreage in GMD3 will reduce the impact of an increasingly dryer climate on the region's economy and create greater stability in the farming sector along with increased employment and more people living in the region.

The two policy mechanisms discussed in this study, 1) senior vs junior water rights and 2) LEMA, represent policy tools that have thus far only been used in Kansas after the the impacts of groundwater depletion have manifested. Thus, they are policy tools for managing a crisis. This begs the question whether one of these conservation enforcement tools or some other policy prescription can be brought to bear to conserve the aquifer before a crisis becomes evident?

Our scenarios demonstrate that any form of conservation enacted today lowers the income of agricultural production in the short and long-term. The differential between the income producers would have earned under the baseline model versus what they are likely to earn if conservation measures were enacted represents an opportunity cost. This opportunity cost is the major obstacle preventing the adoption of any conservation measures.

Policy analysts working for GMD3 in the early 2000s noted this opportunity cost associated with conservation and promoted a crop subsidy to address this differential (Gilson and Aistrup, 2001). What would be the cost if a subsidy were provided for conservation? We estimate this by taking the difference between the average baseline estimated $CF_t$ over the first five years ($420 million) and each scenario's corresponding average $CF_t$. Thus for example, the estimated average annual subsidy to implement the 3X scenario is $113 million and for the 6X scenario is $218 million (2013 dollars). This subsidy is not trivial but, it would allow producers to earn the income they would have if they had continued irrigating at 2013 levels.[10]

Subsidies or other policy interventions need to define an outcome considered desirable to address the current situation. In our case, there are two major goals potential policies could work towards. The first is implementing procedures to extend the current agricultural production regime as long as possible. In this case, addressing the opportunity costs requires an investment (the subsidies) with the expectation that the extended lifetime of the aquifer provides social and economic goods substantial enough to justify the investment before the inevitable pumping reductions imposed by low rates of groundwater recharge.

---

[10]If such subsidies for irrigated crops were provided, policies may be needed to require water savings to be left in the ground in exchange for the crop subsidy. Without such restrictions, research clearly shows that water savings have been used to expand their irrigation operations and maximize profits (Ward and Pulido-Velazquez, 2008a, b; Steward et al., 2013).

This would provide opportunities for local communities to accumulate resources before that happens, relying on the market to determine how much can be accumulated. The second option is using policy tools to navigate the regional economy toward a different agricultural regime, such as a specific dryland agricultural system. In this case, desired outcome would be facilitated by policies more directed toward that outcome, assuming that the region itself would be better off under those conditions.

5 These choices reflect the age old debate in policy making about helping communities accumulate resources to invest in any way they see fit, hoping for a sustainable regional economy to emerge, or provide assistance to move stakeholders along a specific path determined at the regional level. Regardless of which direction policy makers choose, the desired outcome must be clearly defined, as rudderless boats seldom reach their destinations no matter how low the water levels drop.

Perhaps the most important research outcome is that this study establishes the salience of interdisciplinary linked CNH
10 models that seek to untangle and address significant environmental policy issues. Other studies of intensive water use regions have been insightful, but none have incorporated the breadth of our model's components or have used an OpenMI framework. Our modular-holistic model, which includes the major variables of socio-economic impact, crop choice, crop production, and groundwater supply, points toward the policies that can be implemented today to bring a more sustainable future to this region.

Additional research is necessary to refine this CNH model to 1) model the dynamic nature of the grain commodities market,
15 2) take into account new efforts by agra-industry and universities to double grain production levels over the next 15 years, and 3) take advantage of improved scientific models of climate change to more accurately portray the uncertainties that irrigators face and the additional demands for water that climate change may induce in this water challenged region. Researchers in the future can adapt this holistic model to take account of these factors to build new models of sustainability from the wells that pump the water from the aquifer to the communities where people are affected.

20 *Acknowledgements.* This research is funded by a grant from the National Science Foundation (NSF-CNH-0909515).

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

## Appendix A: Supplementary Materials

### A1    Socio-Economic Model

To estimate the population impact of this employment, we conducted a cross sectional time series analysis with panel corrected standard errors (TSCS) controlling for autocorrelation (AR 1) (Beck and Katz, 1995, 2011) from 1970 to 2010, where the time increments are every 5 years (1970, 1975, 1980 etc.), the cross sections are the 12 counties of GMD3, and the error terms are both heteroskedastic and serially correlated. We found:

$$
\begin{aligned}
\text{Total Population}_{i,t} = 1.09 &+ (2.15^{**})\text{Agriculture}_{i,t} \\
&+ (1.33^{**})\text{Manufacturing}_{i,t} \\
&+ (1.45^{**})\text{Construction}_{i,t} \\
&+ (8.14^{**})\text{Health}_{i,t} \\
&+ (3.42^{**})\text{Government}_{i,t} \\
&+ (6.38^{+})\text{Education}_{i,t} \\
&(^{**}p \le .001, {}^{+}p > .05)
\end{aligned}
$$

The manufacturing impact coefficient may seem low, however, most of the workers in meat packing plants are immigrants from Mexico and Central America, who are single or are married and have left their spouses and families in their home country

to work in these meat packing facilities (Broadway and Stull, 2005). Thus, each employee in manufacturing in GMD3 does not yield the type of population impact that manufacturing would in other regions.

## A2   IMPLAN Multipliers for Crops

Table A1 shows IMPLAN multipliers for crops in Western Kansas used in the socio-economic model. Estimating employment impacts can be controversial when computed as a function of total expenditures and revenues, which tends to overestimate the employment impact. We choose instead to calculate employment impacts as a function of $CF_t$ (i.e. profits).

**Table A1.** IMPLAN Multiplier for Crops in Western Kansas

| Irrigated | Direct | Indirect | Induced | Total |
|---|---|---|---|---|
| Total Industry Output | 1.00 | 0.21 | 0.18 | 1.39 |
| Employment | 8.835E-07 | 1.905E-07 | 1.235E-07 | 1.197E-06 |
| Non-Irrigated | Direct | Indirect | Induced | Total |
| Total Industry Output | 1.00 | 0.18 | 0.25 | 1.42 |
| Employment | 8.817E-07 | 1.919E-07 | 1.245E-07 | 1.198E-06 |

## A3   Increasing Crop Efficiency in Southwest Kansas

Table A2 shows the historic data used to estimate the increasing crop yields for corn, soybeans, sorghum, wheat and alfalfa, for both irrigated and dryland varieties.

**Table A2.** Historic Yields in Crop Varieties, Irrigated and Dryland

| | Corn-IRR | Corn-DRY | Soy-IRR | Soy-DRY | Sorghum-IRR | Sorghum-DRY | Wheat-IRR | Wheat-DRY | Alfala-IRR |
|---|---|---|---|---|---|---|---|---|---|
| End year | 2009 | 2009 | 2009 | 2009 | 2009 | 2009 | 2009 | 2009 | 2009 |
| Start year | 1974 | 1974 | 1984 | 1984 | 1974 | 1974 | 1974 | 1974 | 1974 |
| Series length | 35 | 35 | 25 | 25 | 35 | 35 | 35 | 35 | 35 |
| Intercept | 106.86 | 58.84 | 39.934 | 22.938 | 82.766 | 43.294 | 41.95 | 31.455 | 4.307 |
| Slope | 2.489 | 1.109 | 0.5687 | 0.3587 | 0.5442 | 0.8551 | 0.3193 | 0.2139 | 0.0578 |
| End Yield* | 193.96 | 97.658 | 54.152 | 31.906 | 101.81 | 73.222 | 53.126 | 38.942 | 6.33 |
| Percentage** | 0.01283 | 0.01136 | 0.01050 | 0.01124 | 0.00535 | 0.01168 | 0.00601 | 0.00549 | 0.00913 |

*Predicted end year yield ** Annual change/End year yield

Sources: Corn, Soybean, Sorghum, Wheat from Kansas Irrigation Trends (http://www.ksre.ksu.edu/irrigate/OOW/P12/Rogers12Trends.pdf)

Alfalfa from authors analysis of NASS data.

## A4 Coupled Model Prediction Accuracy

Table A3 compares the predictions of regional (GMD3 total) water withdrawals from coupled model simulations from 2013 to 2016, the period of overlap between simulated model results and observed data. Total withdrawals are a key summary measure that combines the results of several model components and drives regional economic impacts. Observed data for years with specific realized weather would not be expected to match mean results, which are averaged across the weather distribution. Observed data diverge from mean predictions by 0.18 to 1.13 standard deviations. Observations from all four years are well within the 90% confidence interval.

**Table A3.** Comparison of Observed and Simulated Water Withdrawals, GMD3 Total, 2013-16

| Year | Observed | Simulated | | |
| | | Mean | Std. Dev. | 90% CI |
| --- | --- | --- | --- | --- |
| | | — billion cubic meters — | | |
| 2013 | 2.478 | 2.334 | 0.3903 | [1.691, 2.976] |
| 2014 | 2.395 | 2.331 | 0.3533 | [1.748, 2.911] |
| 2015 | 1.970 | 2.351 | 0.3357 | [1.799, 2.903] |
| 2016 | 1.970 | 2.378 | 0.3870 | [1.756, 3.000] |

Sources: http://hercules.kgs.ku.edu/geohydro/wimas, model results.