# Peer review of "Conserving the Ogallala Aquifer in Southwest Kansas: From the Wells to People, A Holistic Coupled Natural-Human Model"

_Hydrology and Earth System Sciences, 2017_

## Short Comment (SC1) · 20 Jul 2017

1/19: "Irrigated crops will continue to be crucial for meeting the world's future demands for sustenance" - reword; this essentially says 'will continue to be important for the future' without ever getting around to the present.

2/4: The High Plains Aquifer and the Ogallala Aquifer are not quite the same thing; see bedrock maps available from the USGS, available at https://co.water.usgs.gov/nawqa/hpgw/html/GIS.html. Also, please cite the aquifer boundary: Qi 2010.

2/10: Should also cite publications from Virginia McGuire.

2/19: If you're only studying GMD3, please mention that in the abstract and title. "Sus-

taining the Ogallala Aquifer" strongly implies results that are applicable across the 450,000 sqkm of the High Plains, and GMD3 is not a perfect representation of the whole area (or if you think it is, explain why). The introduction should also give some information specifically about the study area, not just the whole region, and preferably a site map showing the boundaries of the management district (which are not quite the same as the county boundaries OR the aquifer boundaries). People who are familiar with the HPA might not be as familiar with the groundwater management agencies and programs in KS, or might not know the difference between, say, a GMD vs. an IGUCA vs. LEMA. You talk about this later, but it could be introduced early on so that the reader has it in mind.

3: Figure 1 is super fuzzy. Can you get the underlying data and make a new version? I have my own data for this and would be happy to help, but I don't have the KGS data. But I bet Brownie Wilson would send it to you.

4: Figure 2 is great! I'd suggest showing a cone of depression around the well, but it's a really slick conceptual diagram.

5: I recommend reading Hornbeck and Keskin (2014) for another take on the value of the aquifer. Your methods are entirely different, and it would be interesting to think about results of each (yours and theirs) in light of the other.

6/14: You might want to use only the saturated thickness for the area *with enough saturated thickness to support irrigation.* Saturated thickness is heterogeneous within a county, and if you're averaging in a bunch of depleted or non-High Plains area, you'll be underestimating saturated thickness in the areas where there's actual pumping. Also, for a great paper on well yield vs. saturated thickness, try Foster et al. 2015, "Why well yield matters for managing agricultural drought risk."

9/12: Should cite McGuire here.

9/17: "decreases over time"

9/26: "whiskers"

12/20: Again, the average county saturated thicknesses are not really relevant to an individual farmer; it would make more sense to report the average thickness for the area that starts off as irrigable (>9m saturated thickness or so; see Hecox et al. 2002, "Calculation of Yield for High Plains Wells: Relationship between saturated thickness and well yield."

13/9: Again, I recommend Hornbeck and Keskin (2014).

13/15: Whoa, I think you mean the HPA produces 20% of US meat products, but this implies that just GMD3 produces all that meat. For example, when you say "the region" in line 12/16, you're referring to GMD3.

13/20 and elsewhere: "holistic"

Overall, very interesting project and approach. Might want to add some additional language about study limitations, since you're going so far into the future.

—————————————————

---

## Referee Comment (RC1) · Anonymous Referee #1 · 28 Jul 2017

Sustaining the Ogallala Aquifer: From the wells to people, a holistic CNH model, Aistrup et al.

The paper presents a coupled model which is used to analyze different scenarios in the Groundwater Management District 3, of the Ogallala aquifer. The model integrates 4 modules: a socio-economic, crop choice, crop production and a groundwater module. The coupled model is run for 100 years.

It is an interesting model with an interesting application, which relates agricultural policies and its impacts on economy, population and groundwater quantity.

General comments

In the introduction (L13-14, P2), it is mentioned that one of the paper objectives is to

"accurately model the current hyper-extractive CNH system". However there is not a single result that proves that. There is no evaluation of the model performance. Was the coupled model calibrated?

In general the coupled model setup, integration, calibration and limitations are not well explained.

Objective 3 (L16-17, P2) is to "communicate the model's outcomes". There is nothing about this in the paper. Were the model results communicated to the stakeholders? How did the model helped to communicate the policy scenarios?

The title suggests that paper seeks to achieve a sustainable use of the aquifer. A definition of sustainability is shown on page 10, and it is also adapted for the specific situation of the Ogallala aquifer. In L13 P10, it is mentioned "our sustainability policy scenarios focus on maintaining current saturated thicknesses and stemming the current pattern of continuous depletion, while maintaining to the extent possible the employment levels, wealth generation, and population impacts in the region". However, not a single scenario maintains the current saturated thickness. If I understood the model correctly, the objective function seeks to maximize farmers' profits, and it does not have a single constrain on the groundwater level or saturated thickness. The coupled model, as it is, is actually seeking to maintain the profits and employment.

Regarding the coupled model, a diagram showing which data is exchanged between the different models would be very helpful. Did you consider feedbacks between the models?

The data exchange between models is one year, is that enough? What are the dynamics of the groundwater model?

Groundwater model is very simple, how does it impact the results? In Bulatewicz et al., 2010 the authors mentioned that one of the problems in coupled models is that some of their components are very simplified. What are the implication of using a

bass balance equation for modeling the groundwater? What are the implications of the coarse spatial discretization used?

In section 2.6 it is stated that the recharge to the groundwater model is taken from Hansen (1987). I understand that the recharge was not calculated, is that correct? Was it kept constant for the whole 100 years of simulation?

More information about the weather data use should be provided. By how much was the precipitation changed? Was the temperature also changed? By how much? What are the impacts on the recharge?

It is missing a discussion about the uncertainty of the results given by the coupled model. How reliable are they? How accurate can be a 100 years projection? What are the limitations of the model?

Specific comments

Title: Do not use acronyms, if space permits, write: "Coupled Natural Human model"

Title should mentioned that the paper is not analyzing the whole Ogallala aquifer, but the Groundwater Management District 3.

Is it valid to say "Sustaining the Ogallala Aquifer...", when at the end there is no scenario that will do it? (Also see general comment 4).

L7-9, P1: It should be also mentioned the environmental impacts of the expanded presence of irrigated acreage.

L4-7, P2: Nothing is mentioned about the CMD3.

L 16-17, P2: How communicating the model's outcomes will minimize the economic pain for the region's communities? Can you better explain or rephrase?

Improve quality of Figure 1.

Figure 2 is not showing the crop production model.

Figure 2 shows a "Socio-Economic impact model" L1-L8, P5: only mentions a "Economic impact model" are they the same?

L9, P5: The socio-economic impact model described here refers to the "Economic impact model" mentioned in L1-L8, P5?

L23-25, P6: Inside the section "crop choice model", it is mentioned that crop yields were simulated within the model from water response functions. How is this related to the crop yields calculated with the crop production model?

L27-28, P6: It is mentioned that the crop choice model assumes that yields will continue to improve into the future. How is this considered in the crop production model?

L3, P7: Armoa (2015) is not in listed in the references.

L7-8, P9. It is not clear what you mean with "uncertainty by simulating a policy scenario 100 times". Which parameters were modified for making the uncertainty analysis? Where 100 simulations performed for a period of 100 years? Which "policy scenario" was modeled?

Table 2. Show an indicator of the groundwater system.

L11, L13, P13. What do you mean by "policy issues"?

L14, P13. Please specify which are the the "meaningful policies".

Bulatewicz, T., X. Yang, J. M. Peterson, S. Staggenborg, S. M. Welch, and D. R. Steward. "Accessible Integration of Agriculture, Groundwater, and Economic Models Using the Open Modeling Interface (OpenMI): Methodology and Initial Results." Hydrol. Earth Syst. Sci. 14, no. 3 (March 16, 2010): 521–34. doi:10.5194/hess-14-521-2010.

---

## Referee Comment (RC2) · Anonymous Referee #2 · 8 Aug 2017

[revised manuscript text omitted]

July 31, 2017

Summary:

This paper investigates impacts of a range of water policy proposals on sustaining the Ogallala Aquifer in Kansas, USA.  It integrates irrigation water and land use patterns along with climate and economic, and population forecasts.  It examines several future outcomes of policy scenarios transitioning the current groundwater-based economic system toward more sustainable paths for the social, economic, and natural components of the integrated system.  Results show that increased irrigated acreage will reduce the impact of an increasingly dryer climate on the region's economy and create greater long-term stability in the farming sector along with increased employment and population in the region.

Overview Comments:

The paper presents a comprehensive multidisciplinary linked model of the hydrology, agronomy, economics, and policies associated with groundwater pumping in the Ogallala Aquifer.  It's a nice piece of work.  Still, I would like the authors to describe their conclusions on what they found could promote a more sustainable groundwater use pattern compared to the current groundwater depletion pattern in their study region.  The paper described several policy proposals, but I found nothing in the conclusions stating which policies were the most sustainable for their definition of sustainability.

Technical comments:

Figure 2: A flow chart would help in describing the moving parts of the model to complement the figure.

Page 6, line 10:  Does the PMP model calibrate to a base year or base condition?

Page 6, line 12: Is the farm income optimization model a single year at a time?  If so, why isn't a longer time horizon used for farmers.

Page 6, line 30:  Do the increased yields require additional water, common in many crop water use models, for which yields are typically proportional to water use?

Page 8, line 28:  how is kriging different from regression plus simple smoothing?

Page 9, line 11:  How is sustainability measured to justify the statement?

Page 10, line 11:  Is water importation into the Ogallala study region allowed to offset aquifer declines?

Page 11, line 12:  Discussion on LEMA:  A large group of junior water right holders could outvote senior water right holders, for which the majority vote would require all irrigators to reduce use by a set proportion.  But the junior water right holders arguably don't have a right to vote, as their junior status gives them little to no pumping right in a shortfall period.

Page 12 line 1:  Why are the first two policy options not sustainable?  Is there a measure of sustainability defined somewhere?

Overall:  I have also attached a few comments and questions in the blue diamonds in the attached manuscript, mostly of a minor editorial nature.

---

## Author Comment (AC1) · 1 Sep 2017

We concur with comment 1/19, and we will revise accordingly

2/14 The High Plains Aquifer is used for the Ogallala Aquifer deposits and some of the related units, but High Plains is more universally used. We will specifically state this in the revised manuscript.

2/10 We can add McGuire cites easily.

2/19 Adding Groundwater Management District #3 to abstract is a good idea. Also providing a better sense of the generalizability of our study to the aquifer is a good suggestion that we can easily do.

[Figure]

3. We are willing create or obtain a new Figure 1, or revise figure 1 based on the editorial policies of HESS.

4. Figure 2, we will add a cone of depression.

5. We concur. We believe Hornbeck and Foster's study should be worked into the text.

6/14: We will look at the piece by Foster et al (2015) within the context of our water model.

9/12: We will cite McQuire.

9/17: We concur.

9/26: Yes, we will revise.

12/20: Yes, an individual irrigator is interested in how much water is in their particular well(s) to support their crop in their fields. Yet, it is important for overall management and prediction of net benefit to society through groundwater extractions to integrate the volume of water over a study region. For our study, we decided to integrate to a common scale that is consistent across all model components. And, a scale important for predicting crop production and the capacity to sustain the important rural communities into the future. We will make sure this is clear in the revision.

13/9: We will cite Hornbeck and Keskin

13/15: Our statistics are correct.

13/20: Yes.

We agree to add some language about the limitations of projections that are 100 years into the future.

---

## Author Comment (AC2) · 1 Sep 2017

The general comments 1, 2, 3, 4, and 5 deal with the model set up, depiction, calibration, communication of outcomes, and developing a sustainable policy option.

First, in our revision, we are more than happy to provide a diagram (GC5) depicting the data exchanges and feedback loop. This diagram should also help to better explain the model setup (GC2) and facilitate a discussion of the limitations of this 100-year predictive model, the accuracy of which decreases overtime.

Second, each separate component of the coupled model is calibrated independently prior to being linked to the other components. Citations are provided in the text and will not be repeated in this response. However, AR1 indicates that we did not provide

an evaluation of the coupled model's performance (GC1). At one level, we disagree with this assessment because we calibrated each individual disciplinary model prior to linking it to the next. We nonetheless recognize the burden of proof is on us to demonstrate that once the model is coupled, the coupled model produces outcomes that are consistent with existing data. In the revision, we plan to show predicted outcomes for the period between 2000 and 2014 and a comparison of the model's outcomes with the published data in a supplementary section for the paper. Communicating the model's outcomes (GC3) to stakeholders and policy makers will be partially accomplished by this peer reviewed publication process. After our peers have evaluated this model and found it acceptable, additional steps are planned to communicate the model's outcome to other policy-makers and stakeholders in the Groundwater Management District 3 study region.

The coupled-model is robust enough to develop a sustainable outcome for the aquifer (GC4), which we define as halting the current trajectory toward depletion. However, this policy option is not evaluated, since it is not under consideration at this time within the study region. This study advances the societal need to understand the ramifications of coupled natural/human systems, while contributing towards a more sustainable dialog grounded in real-world possibilities.

GC6: Groundwater pumping is based upon an annual decision for when to start pumping and when to stop, where the well is traditionally left on throughout the growing season. Thus, the dynamics are drawdown throughout a growing season and recovery before the next pumping cycle, where large drawdowns may occur when the wells actively pump (Mullican,2012), and a new elevation becomes established during a recovery period (Duganet al., 1994, p. 23). Thus, the groundwater model exchanges data at the same annual frequency as the economic crop choice model that dictates the annual water requirements.

GC7: The baseline model predictions accurately reproduced the groundwater data throughout the historical period. Model results were compared to the future predictions

of a higher-resolution fishnet model of Seward County from Steward et. al (2009). Groundwater levels were consistent between studies. Similarly, the results from the model resolution of this study, when aggregated to the region, reproduce longer term regional projections (Steward et al. 2013).

GC8: The recharge component was kept consistent throughout this study. Given the relatively thick soil units throughout much of the region (Gutentag et al. 1984B), recharge rates are low and may take decades or longer to reach saturated groundwater (McMahon et al. 2007). This is consistent with Sophocleous [2005] who noted that "Groundwater pumping usually has little impact on the recharge".

McMahon PB, Dennehey KF, Bruce BW, Gurdak JJ, Qi SL (2007) Water-Quality Assessment of the High Plains Aquifer, 1999 – 2004 (US Geological Survey, Reston, VA), Professional Paper 174.

GC9: AR1 wants some details added to section 2.5 on what the resulting dryer-year resampling of years looked like statistically. We will compare the statistics of the original 27 years of weather data and compare them to a set where the dryer years are 25% more likely.

GC10: We will add standard deviations to table 1.

AR1 also made a number of useful "specific comment" (SC). We agree with SC 1 through 10, 13, and 15. We will make the suggested revisions.

SC 11 and 12 ask questions regarding the crop production model. The crop choice model is not calibrated to a specific year, rather to the mean outcomes during the base period. Calibrating to a specific year will "overfit" the model so that it replicates that year exactly but likely at the cost of very poor prediction accuracy for other years. The mean approach doesn't predict any single year exactly, but makes the model provide a better match with the range of observations during the base period.

SC 14 notes a poor choice of wording on our part. We will revise our text to reflect

that we ran all of the scenario simulations 100 times, where each policy scenario simulated a period of 100 years, and that the parameter we modified was the weather-year resampling.

SC 16 and 17 draws attention to the conclusion where we use the term "policy issues" and "meaningful policies." We use the term policy to refer to a course or principle of action adopted or proposed by a government/regulatory agency. We will delete the words "issues" and "meaningful".
* * *

---

## Author Comment (AC3) · 1 Sep 2017

AR 2 notes that s/he would like for the conclusion to focus on what policies could be adopted to promote a more sustainable groundwater use pattern compared to the status quo. We concur and will redraft the conclusion to outline such a policy.

AR2, Technical comments

As noted above, we agree to add a diagram.

Page 6, lines 10 and 12: The crop choice model was fitted to data over a span of years, rather than a single base year (also see under AR1: SC11 and 12).

Page 6, line 30 As the reviewer notes, it is not uncommon for crop models to represent

yield as being proportional to water use. Fortunately, this is not the case with EPIC. Except, of course, when water is limiting, the plant biomass accumulated in each time interval is related to the intercepted solar radiation, a procedure called the "radiation use efficiency" approach. Then yield is estimated as a fraction (the "harvest index") of biomass. So water use is related to biomass, rather than the other way around.

Page 8, line 28 Replacing kriging with interpolation is fine.

Page 9, line 11 Yes, provided on page 10, lines 5 through 15. The definition is adapted from Peter Gleick and stated in the paper: "maintaining current saturated thicknesses and stemming the current pattern of continuous depletion, while maintaining to the extent possible the employment levels, wealth generation, and population impacts in the region."

Page 11, line 12 A LEMA does not work this way. The LEMA is designed to develop consensus at the grassroots, such that most if not all contiguous water rights holders, Junior and Senior, would agree to the same water withdrawal policy.

Page 12, line 1 The first two policy options are not sustainable because the levels of water savings are not enough to reverse the current pattern of continuous depletion. (See definition of sustainability, above).

We concur with AR2's editorial suggestions.

---

## Author Response (AR1)

Dear Editor,

We made the following revisions:

Revised figures 1 and 2.  We added a flow diagram depicting data exchanges to figure 2.

Added verbiage regarding the limitations of a 100 year predictive model, the accuracy of which decreases overtime.

Revised the title and language in the text to reflect that 1) our scenarios do not obtain sustainability, as we defined it, but do help to "conserve" the aquifer as a resource for future generations.  We also made numerous smaller edits, all consistent with the comments of the reviewers.

We included in the text or added as a footnote the following clarifications based on the reviewers' comments:

- Groundwater pumping is based upon an annual decision for when to start pumping and when to stop, where the well is traditionally left on throughout the growing season. Thus the dynamics are drawdown throughout a growing season and recovery before the next pumping cycle, where large drawdowns may occur when the wells actively pump (Mullican,2012), and a new elevation becomes established during a recovery period (Dugan al., 1994, p. 23).  Thus, the groundwater model exchanges data at the same annual frequency as the economic crop choice model that dictates the annual water requirements.
- The baseline model predictions accurately reproduced the groundwater data throughout the historical period.  Model results were also compared to the future predictions of a higher-resolution fishnet model of Seward County from Steward et al. 2009. Groundwater levels were consistent between studies.  Similarly, the results from the model resolution of this study, when aggregated to the region, reproduce longer term regional projections (Steward et al. 2013).
- The recharge component was kept consistent throughout this study.  Given the relatively thick soil units throughout much of the region, Gutentag et al. (1984B), recharge rates are low and may take decades or longer to reach saturated groundwater [McMahon et al. 2007]. This is consistent with Sophocleous [2005] who noted that "Groundwater pumping usually has little impact on the recharge".
- Details on what the resulting dryer-year resampling of years looked like statistically. We compared the statistics of the original 27 years of weather data and compare them to a set where the dryer years are 25% more likely.
- Revised table 1, adding aquifer status variable and standard deviations.
- Explained that the crop choice model is not calibrated to a specific year, rather to the mean outcomes during the base period. Calibrating to a specific year will "overfit" the

model so that it replicates that year exactly but could do a terrible job with the year before or after. The mean approach doesn't give us an exact fit for any single year but makes the model match better with the data cloud.

- Revised the conclusion to focus on which policies could be adopted to promote a more sustainable groundwater use pattern compared to the status quo. We discussed the cost of subsidies for irrigators for conservation and the need for the GMD3 to determine its goals for the future.
- Noted that even though it is not uncommon for crop models to represent yield as being proportional to water use, that this is not the case with EPIC. Except, of course, when water is limiting, the plant biomass accumulated in each time interval is related to the intercepted solar radiation, a procedure called the "radiation use efficiency" approach. Then yield is estimated as a fraction (the "harvest index") of biomass. So water use is related to biomass, rather than the other way around.
- Change the language associated the explanation of LEMAs. Majority rule does not apply to LEMAs, rather it is about consensus building.
- Added citations for McGuire, Foster et al., and Koehn.

As an additional check on the validity of our coupled system, we compared published groundwater use and saturated thickness data with our model from 2013 to 2016. We found that our model accurately predicted the declines in groundwater statistics.

We hope you find our MS acceptable.

Best wishes,

[revised manuscript text omitted]